# Testing the existence of an unadmixed ancestor from a specific population t generations ago

Gabriel Illanes[1]*, María Inés Fariello[2,3], Lucía Spangenberg[2,5], Ernesto Mordecki[1], Hugo Naya[2,4]

1 Centro de Matemática, Facultad de Ciencias, Universidad de la República, Montevideo, Uruguay, 2 Bioinformatics Unit, Institut Pasteur de Montevideo, Montevideo, Uruguay, 3 Facultad de Ingeniería, Universidad de la República, Uruguay, 4 Facultad de Agronomía, Universidad de la República, Montevideo, Uruguay, 5 Departamento de Informática, Universidad Católica del Uruguay, Montevideo, Uruguay

* gillanes@cmat.edu.uy

## Abstract

The ancestry of each locus of the genome can be estimated (local ancestry) based on sequencing or genotyping information together with reference panels of ancestral source populations. The length of those ancestry-specific genomic segments are commonly used to understand migration waves and admixture events. In short time scales, it is often of interest to determine the existence of the most recent unadmixed ancestor from a specific population t generations ago. We built a hypothesis test to determine if an individual has an ancestor belonging to a target ancestral population t generations ago based on these lengths of the ancestry-specific segments at an individual level. We applied this test on a data set that includes 20 Uruguayan admixed individuals to estimate for each one how many generations ago the most recent indigenous ancestor lived. As this method tests each individual separately, it is particularly suited to small sample sizes, such as our study or ancient genome samples.

## 1 Introduction

The information that we have about our ancestors and their origins comes mostly from our families' stories and in best cases from reconstructed genealogies. With genomic information we can build more precise and reliable genealogies and even calculate the proportions of our ancestries. For instance, there can be a discordance between the self-declared ancestry and the ancestry estimated from a genome study, as shown in [1]. Even if the exact genealogy of an individual is known, it is very difficult to estimate the proportion of a particular ethnic group, since the amount of genetic material that is yielded from one generation to the next one is highly variable [2, 3]. As an example, according to Coop's calculations using simulations, with a high probability (close to 1) one inherits almost zero genomic material from at least an ancestor that lived only 7 generations ago.

Data Availability Statement: All the data and code that was used in this work was uploaded to figshare. Data: https://figshare.com/articles/dataset/Testing_the_existence_of_an_unadmixed_ancestor_from_a_specific_population_t_

generations_ago/20277795 Code: https://figshare.
com/articles/software/Testing_the_existence_of_
an_unadmixed_ancestor_from_a_specific_
population_t_generations_ago_code_/20277813.

**Funding:** Gabriel Illanes acknowledges support of
Agencia Nacional de Investigación e Innivación
(ANII-Uruguay) and Comisión Académica de
Posgrado (CAP-Udelar) The Urugenomes project
was funded by BID (Banco Iberomericano de
desarrollo) Proyecto ATN / KK-L4584-JR
"Fortalecimiento de las capacidades técnicas y
humanas para las exportaciones de servicios
genómicos". Additionally, Maria Ines Fariello and
Lucia Spangenberg obtained partial support from
the ANII-Uruguay FSDA 1 2017 1 143647 and
Lucia Spangenberg and Hugo Naya are also
supported by FOCEM (MERCOSUR Structural
Convergence Fund).

**Competing interests:** The authors have declared
that no competing interests exist.

Genomic data has been used to study a great variety of human population characteristics, such as population structure [4], admixture events [5–7] and the estimation of ancestry proportions from an individual's genome [8]. In particular, the challenge of estimating the local ancestry (ancestry-specific genomic segments) of an admixed individual, which means to determine the tracts in the genome corresponding to different ancestral populations (eg. European, African and Native American), has been successfully addressed with different approaches [9–11]. The possibility of representing the genome by a disjoint series of tracts with different ancestries enables the application of mathematical modeling tools to retrieve interesting information regarding the history of an individual. For instance, given a particular pedigree one can model admixture events by stochastic processes, allowing the study of inference methods for admixture deconvolution and segregation of tracts in the pedigree. Assuming that the target tracts are rare, hence they are unlikely to recombine, an admixture tract-length distribution was derived in [5]. Furthermore, some model assumptions were relaxed in [7], modeling tracts that descended from multiple migrant ancestors under a simplified model (Markovian Wright–Fisher). Additionally, a dyadic interval-based stochastic process for generating admixture tracts was developed by [6].

Here, we have developed a hypothesis test to assess whether it is likely that one of the individual's ancestors $t$ generations ago was an unadmixed ancestor (e.g. complete individuals genome with only one ancestry), given a fixed number $t$ of generations and the length of the ancestry-specific tracts for every autosome.

We applied this test on a data set that includes the genomes of 20 Uruguayan individuals (ten descendants of the past local indigenous groups [1] and ten afro-descendants). According to historical records, most Uruguayan Amerindian were exterminated in 1831 [12]. Only some of them survived, and several women and children were taken as prisoners. As far as we know, no unadmixed indigenous individuals are living nowadays among the Uruguayan population. Our previous study has shown that there is non-negligible indigenous ancestry in this particular data set, indigenous percentages range from 7% to almost 40%. Also, mitochondrial DNA haplogroups show indigenous haplogroups such as B or C. Admixture results together with admixture graphs show a genomic affinity with Amazonian and one Andean indigenous group [1].

In the current study we want to find a new hypothesis test that brings deeper information about each individual's history. In this sense, our motivation relies on knowing whether descendants of the indigenous groups had ancestors that survived the genocide, i.e. had an unadmixed ("complete") indigenous ancestor about just 3 or 4 generations ago; or that the admixture events occurred before the genocide, meaning that those few survivors were admixed with the general population.

## 2 Methods

### 2.1 Definitions and notations

Let $a^0$ be an individual, and $\mathbf{a}^t = \{a_1^t, \ldots, a_{2^t}^t\}$ the individual's ancestors $t$ generations ago. The individuals $a_{2i-1}^t, a_{2i}^t$ mate, with offspring $a_i^{t-1} \in \mathbf{a}^{t-1}$, for all $i$ and $t$. We will assume that $a_0$ and all their ancestors up to generation $t$ are admixed with respect to a family of ancestral populations $\{\mathcal{P}_\lambda\}_{\lambda \in \Lambda}$. Furthermore, for a given individual $a_0$, let $c_i$, $i = 1, \ldots, 22$ denote their autosomes. Each $c_i$ consists of a chromosome pair $(c_i^1, c_i^2)$.

In this work, we will use the terms "segment" and "tract" of a chromosome in the following way.

**Definition 2.1** (Segment of a chromosome). *We will refer to a haplotype of a chromosome* $[x_1, x_2] \subset c_i^j$ *as a "segment" (it can contain genetic information related to different ancestral populations).*

**Definition 2.2** (Tract of a chromosome). *We will refer to a segment with all its genetic information related to the same ancestral population* $\mathcal{P}_\lambda$ *as a "$\mathcal{P}_\lambda$ ancestral tract", or just "$\mathcal{P}_\lambda$-tract".*

This distinction is arbitrary, but it is important to note if a given segment has all its genetic information related to the same ancestral population or not.

In this work, we will measure lengths of segments and tracts in Morgans, which is a usual measure unit to consider, and very suited for this work.

**Definition 2.3** (Morgan). *A "Morgan" is defined as the distance between chromosome positions for which the expected number of recombinations between homologous chromosomes in a single generation is 1.*

In this work, we will assume that, for a given individual, every chromosome can be considered as a concatenation of tracts:

$$c_i^j = h_1^{i,j} \ldots h_n^{i,j}$$

where each $h_k^{i,j}$ is a $\mathcal{P}_\lambda$-tract, for some $\lambda \in \Lambda$. We will also assume that, for a given individual $a_0$, we know the length (in Morgans) and ancestral population related to every tract $h_k^{i,j}$, for every chromosome of $a_0$.

**Definition 2.4** ($\mathcal{P}_\lambda$-complete individual). *For a given* $\lambda \in \Lambda$, *we say that an individual a is* $\mathcal{P}_\lambda$-complete *if all their chromosomes are* $\mathcal{P}_\lambda$-tracts.

## 2.2 The hypothesis test

The objective of this work is, for a fixed generation $t$, develop a hypothesis test to assess if at least one of the ancestors $a_1^t, \ldots, a_{2^t}^t$ is $\mathcal{P}_\lambda$-complete, for a given $\lambda \in \Lambda$.

Without loss of generality, let us focus on the two population case, $\mathcal{P}_1$ and $\mathcal{P}_2$. For a given $t$, we are interested in doing the following test,

$H_0$:   $a^0$ has, at least, one $\mathcal{P}_1$-complete ancestor, $t$ generations ago.

$H_1$:   $a^0$ has no $\mathcal{P}_1$-complete ancestors, $t$ generations ago. $\qquad\qquad$ (1)

One of our major problems is not having information about an individual's ancestors $t$ generations ago. If we would like to sample the $2^t$ ancestors of $a^0$, $t$ generations ago, we would not have enough information about $a^0$, or about their ancestors, that we can use to fix a realistic distribution function on the space of all possibilities. Our strategy, then, is to focus on a case of $H_0$, where we can fix the ancestors' pedigree.

$\bar{H}_0$:   $a^0$ has exactly one $\mathcal{P}_1$-complete ancestor, $t$ generations ago.

$\qquad\quad$ The other ancestors are $\mathcal{P}_2$-complete. $\qquad\qquad\qquad\qquad\qquad$ (2)

$H_1$:   $a^0$ has no $\mathcal{P}_1$-complete ancestors, $t$ generations ago.

Without loss of generality, if $a^t \in H_0$ or if $a^t \in \bar{H}_0$, we assume that $a_1^t$ is the $\mathcal{P}_1$-complete ancestor of $a^0$ (if not, reorder the family tree to make it so). While there are several distribution functions supported in $H_0$ which could be used to sample $a^t$, there is only one possibility in $\bar{X}_0$; that is, $a_1^t$ is a $\mathcal{P}_1$-complete ancestor, and $a_2^t, \ldots, a_{2^t}^t$ are all $\mathcal{P}_2$-complete ancestors. The test 1 is a composite hypothesis test, whereas the test 2 is a simple hypothesis test. In the S1 File, we

show that we can build statistics such that their $p$-value under $H_0$ is always stochastically smaller than their $p$-value under $\bar{H}_0$.

## 2.3 Mathematical model

We assume that chromosomes can be thought as real intervals, instead of a sequence of bases. This assumption aims to ease some computational burden (we will explore the need for simulations in subsection 2.5). If this assumption is not made, we have to consider a large amount of very long vectors during the simulations, which would consume a lot of computational resources. When this assumption is made, we model each chromosome as an interval, and simulate the Poisson process in the interval using the exponential distribution. This assumption is not a strong one, because the number of recombination points introduced during the meiosis is much smaller than the the total number of bases on each chromosome.

For a given ancestor $a_k^t$ and chromosome $i$, we consider the chromosome pair $(c_i^1, c_i^2)$. During the meiosis, those chromosomes recombine to create an offspring chromosome as follows:

1. Recombination points are introduced using a Poisson process with parameter $L_i$ (length of the chromosome in Morgans). Including the borders of the interval $[0, L_i]$, we obtain $\{x_0 = 0, x_1, \ldots, x_n, x_{n+1} = L_i\}$.

2. A parent chromosome is selected randomly. The segment $tr_1 = [0, x_1]$ in the selected chromosome will be the first segment of the offspring chromosome.

3. At the point $x_1$, switch to the other chromosome, and concatenate the segment $tr_2 = [x_1, x_2]$.

4. The process is repeated until the length of the offspring chromosome is $L_i$.

This process is illustrated in Fig 1 (meiosis for complete chromosomes and meiosis for admixed chromosomes).

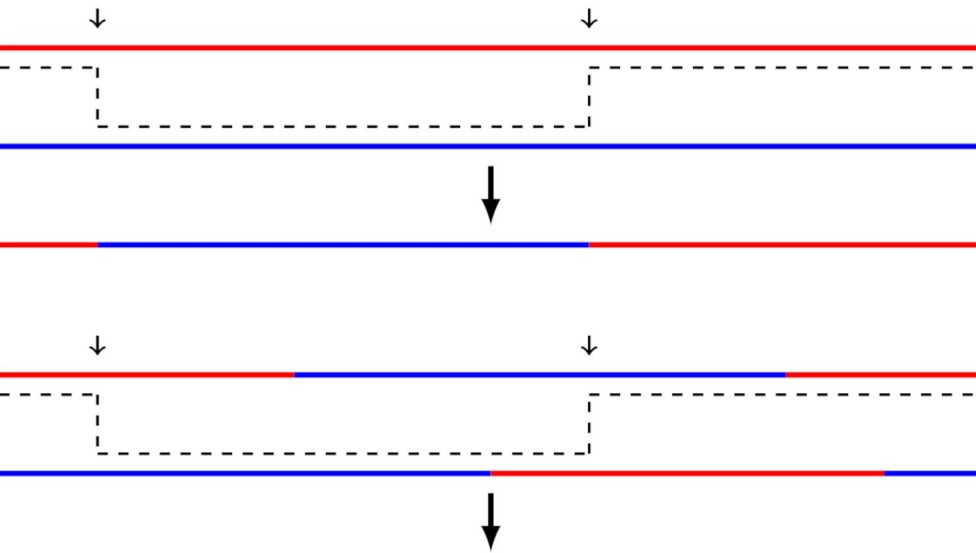

**Fig 1.** Top: recombination of two $\mathcal{P}_\lambda$-complete chromosomes during meiosis to create an admixed offspring. The first one is $\mathcal{P}_1$-complete (red) and the second one is $\mathcal{P}_2$-complete (blue). Bottom: Two admixed chromosomes consisting of $\mathcal{P}_1$-tracts (red) and $\mathcal{P}_2$-tracts (blue) recombine during meiosis to create an offspring chromosome.

Alternatively, we could have chosen a Wright-Fisher model [5, 7] as our model. As it possess the Markov property, it is easier to develop mathematical models and tests; however, it fails to capture some structures when we work at an individual level, with small values of $t$. One such example is when a $\mathcal{P}_1$-complete mates with a $\mathcal{P}_2$-complete individual: under the Diploid Wright-Fisher model, one of the offspring chromosomes will be $\mathcal{P}_1$-complete and the other one will be $\mathcal{P}_2$-complete; whereas in a Markovian Wright-Fisher model, both offspring chromosomes can be admixed, or even lose the genetic information of one of the parents. We conclude that Markovian Wright-Fisher models are only suitable when working with whole populations and large values for $t$.

### 2.4 Definition of the test statistic

Let us fix the parameter $t$ of our hypothesis test; the objects we define in this section depend on $t$, but we will not index it to simplify the notation. Our objective is to define a test statistic for the test 2 that bounds the same test statistic for the test 1. Our strategy is to do that in two steps: first, we will construct a score for each chromosome pair; and second, we will combine all those scores into a test statistic in different manners.

**2.4.1 Chromosome scores.** **Definition 2.5** (Chromosome statistics). *Given a chromosome pair, we will consider two possible statistics*:

- $m_i^{max}$ *is the maximum length among all $\mathcal{P}_1$-tracts in the chromosome pair,*

- $m_i^{sum}$ *is the maximum sum of lengths of all $\mathcal{P}_1$-tracts among the chromosome pair.*

In order to ease the notation, we will denote the chosen statistic by $m_i$, for $i = 1, \ldots, 22$, unless the distinction is needed.

As the lengths of the chromosomes are all different, the $m_i$ are not comparable across chromosome pairs. Let $M_i$ be the random variable from which $m_i$ is sampled, and define $p_i$ as:

$$p_i = \mathbb{P}_{H_0}(M_i \leq m_i \,|\, M_i > 0), \tag{3}$$

the probability under $H_0$ of observing a smaller chromosome statistic $M_i$ that the one observed $m_i$. As a technical consideration, we will condition the probability $p_i$ to $M_i > 0$ to improve the performance of the hypothesis test (we refer to the supplementary section for further details).

**Definition 2.6** (Chromosome scores). *Let $m_i$, for $i = 1, \ldots, 22$ be a chromosome statistic. Denote $M_i$ the random variable from which $m_i$ is sampled. For $i = 1, \ldots, 22$, we define the chromosome score $p_i$ as*

$$p_i = \mathbb{P}_{H_0}(M_i \leq m_i \,|\, M_i > 0),$$

*It is important to note that $\mathbb{P}_{H_0}$ depends on $t$.*

**The chromosome scores have similar distributions across all chromosomes, and thus we can compare and combine them**: To see that, let us denote as $P_i$ the random variable used to sample $p_i$. Using the probability integral transform, and observing that $w_0 = \mathbb{P}_{H_0}(M_i = 0) \neq 0$ and $w_L = \mathbb{P}_{H_0}(M_i = L_i) \neq 0$, we deduce the distribution function of the random variable $P_i$:

$$P_i \overset{d}{=} \mathbf{1}_{\{V > 1 - w_L\}} + U\mathbf{1}_{\{w_0 \leq V \leq 1 - w_L\}}, \text{ where}$$
$$U, V \sim \text{Unif}(0, 1) \tag{4}$$
$$U, V \text{ are independent}$$

where $\overset{d}{=}$ denotes that the random variables are equal in distribution. We observe that

$\mathbb{P}_{H_0}(M_i = 0) \neq \mathbb{P}_{H_0}(M_j = 0)$ and $\mathbb{P}_{H_0}(M_i = L_i) \neq \mathbb{P}_{H_0}(M_j = L_j)$ if $i \neq j$, but we will avoid the chromosome indexation to simplify the notation. We conclude that $P_i \neq P_j$ if $i \neq j$, but they have the same range, and they both behave as uniform distributions in the interval $(0, 1)$. They only differ in the weight of their atoms (when $M = 0$ or $M = L$).

**2.4.2 Combining the chromosome scores into a test statistic.** The distribution of the random variable $P_i$ is given by Eq 4, and we observe that $P_i \overset{d}{\neq} P_j$ if $i \neq j$. Assuming that the recombination spots are independent between chromosomes, then all $M_i$ are independent, and thus the $P_i$ are independent.

The distribution function of $M_i$ under $H_0$ is unknown, hence to compute the $p_i$s we can simulate using Monte Carlo their distribution under $\bar{H}_0$. The probabilities we need to approximate are $p_{H_0}(M_i \leq m_i)$, $w_0$ and $w_L$, that are enough to compute $p_i$ and approximate their theoretical distribution function.

Subsequently, we propose two different ways of combining all chromosome scores $p_i$ into a test statistic whose $p$-value is easy to compute. Our first proposal is to define $p_{max}$ as the maximum of all $p_i$

$$p_{max} = \max_{i=1,\dots,22} p_i. \tag{5}$$

If $F_i$ is the distribution of $P_i$, then the distribution of the random variable $P_{max}$:

$$F_{P_{max}}(x) = \prod_{i=1}^{22} F_i(x), \ \forall x \in [0, 1], \tag{6}$$

and the final $p$-value as $p = F_{P_{max}}(p_{max})$.

The second idea is to consider $p_{sum}$, the sum of all chromosome scores

$$p_{sum} = \sum_{i=1}^{22} p_i. \tag{7}$$

As the distribution functions of all $P_i$, $F_i$, are different, it is very difficult to compute the theoretical distribution $F_{P_{sum}}$. However, as we know $F_i$ for all $i$, it is easy to approximate the $p$-value of the test using Monte Carlo simulations.

Considering that both $p_{max}$ and $p_{sum}$ can be constructed using both definitions of $m_i$ defined in 2.5, we propose four variants of the hypothesis test 2.

**Definition 2.7**. *We define the following test statistics.*

- $\boldsymbol{p}_{mm}$ *when we use* $m_i^{max}$ *and we consider the maximum of all* $p_i$.

- $\boldsymbol{p}_{sm}$ *when we use* $m_i^{sum}$ *and we consider the maximum of all* $p_i$.

- $\boldsymbol{p}_{ms}$ *when we use* $m_i^{max}$ *and we consider the sum of all* $p_i$.

- $\boldsymbol{p}_{ss}$ *when we use* $m_i^{sum}$ *and we consider the sum of all* $p_i$.

As we are interested in the hypothesis test 1, we need the following theorem 2.1, that allows us to simulate under $\bar{H}_0$ and use the results to bound the $p$-values of test 1.

**Theorem 2.1**. *Let p be any of the four p-values defined for the hypothesis test 2 ($p_{mm}$, $p_{sm}$, $p_{ms}$, $p_{ss}$). Then, CR = $\{p \leq \alpha\}$ is a critical region for the test 1 with probability $\beta \leq \alpha$.*

In other words, we can control the type I error of the hypothesis test 1 by controlling the type I error of the hypothesis test 2. The proof of theorem 2.1 is detailed in the S1 File.

Algorithm 1 shows a summary of the methodology we developed. Usually, one should consider all choices for chromosome statistic and test statistic. As long as they result in few tests

(i.e. we avoid multiple testing issues), we can consider only the smallest $p$-value obtained, and reject the null hypothesis if any of the test rejects.

**Algorithm 1** Ancestry test algorithm for a given individual

```
1) Set an objective ancestral population and a number of generations t.
2) Choose a chromosome statistic; usually mᵢᵐᵃˣ or mᵢˢᵘᵐ. Compute all 22
   statistics.
3) Using simulations, estimate the distribution functions for the
   chromosome statistics under H̄₀, and compute the chromosome scores.
4) Choose a way of combining all chromosome scores into a test statis-
   tic; usually the sum of scores pˢᵘᵐ, or the maximum of scores pᵐᵃˣ.
5) Compute the distribution of the test statistic, and compute the
   test p-value p.
return p
```

## 2.5 Theoretical computation of the distribution of a chromosome pair

The objective of this section is to show that the computation of the chromosome statistics distribution, under $\bar{H}_0$, is a very difficult problem. The main reason is that our model for chromosome recombination is not a Markov process when we condition only to the genetic information of the parent chromosomes.

An important observation is that, under $\bar{H}_0$ and for a given $t$, one of the chromosomes in each chromosome pair $(c_i^1, c_i^2)$ of $a_0$ will be a $\mathcal{P}_2$-complete chromosome. Let us assume $c_i^2$ is the $\mathcal{P}_2$-complete chromosome. We only need to focus on the $\mathcal{P}_1$-tracts in $c_i^1$, and deduce the distribution function of the chosen chromosome pair statistic $m_i$.

Let us start for $t = 1$. In this case, we have two chromosome pairs (one for each parent). One of the chromosome pair is $\mathcal{P}_1$-complete, and the other one is $\mathcal{P}_2$-complete. The first chromosome pair recombines to create a $\mathcal{P}_1$-complete chromosome, and the other pair recombines to create a $\mathcal{P}_2$-complete chromosome (as expected). Thus, $c_i^1$ is a $\mathcal{P}_1$-complete chromosome, so we conclude that $\bar{H}_0$ is false if neither of the chromosomes is $\mathcal{P}_1$-complete.

For $t = 2$, $c_i^1$ will be a recombination of a $\mathcal{P}_1$-complete chromosome and a $\mathcal{P}_2$-complete chromosome (as in Fig 1). We observe that the length of each tract is distributed as $exp(1)$, and tracts alternate between $\mathcal{P}_1$ and $\mathcal{P}_2$. Let $N_i$ be the amount of $\mathcal{P}_1$-tracts in $c_i^1$. If we can compute the distribution of $N_i$, we will be able to compute the distribution of $m_i$, whichever we choose. Let $T_1, \ldots, T_{N_i}$ be the lengths of the $\mathcal{P}_1$-tracts in $c_i^1$, then the distribution functions for $m_i^{max}$ and $m_i^{sum}$ are

$$F_i(m_i^{max}) = \sum_{j=0}^{+\infty} \Pr(N_i = j)\Pr(\max\{T_1, \ldots, T_j\} \leq m_i^{max}), \qquad (8)$$

$$F_i(m_i^{sum}) = \sum_{j=0}^{+\infty} \Pr(N_i = j)\Pr(\sum_{k=1}^{j} T_k \leq m_i^{sum}). \qquad (9)$$

We conclude that, for $t = 2$, the distributions can be computed, or at least approximated with precision. However, for $t \geq 3$, it is not clear how to compute the distribution of the lengths of the $\mathcal{P}_2$-tracts. The problem is that, after we reach a recombination point in $c_i^1$, we can not compute the exact probability of the next tract being $\mathcal{P}_1$ or $\mathcal{P}_2$, because it is not a Markov process. This means that we can not compute the distribution of $N_i$, and can not recover the Eqs 8 and 9.

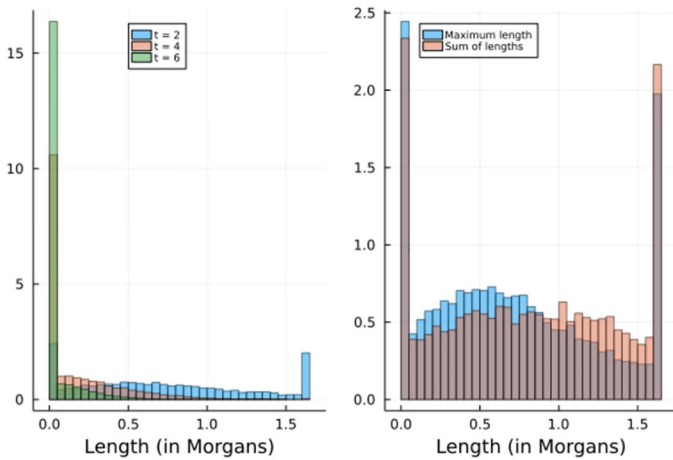

**Fig 2.** Left: Histogram of 10000 simulations of $m_{11}^{max}$ for generations $t = 2, 4, 6$ under $\bar{H}_0$. We can observe that the atom in 0 becomes larger and, in general, the distributions of $m_{11}^{max}$ become stochastically smaller as $t$ increases. Right: Histogram of 10000 simulations of $m_{11}^{max}$ and $m_{11}^{sum}$ for generation $t = 2$ under $\bar{H}_0$. The distribution of $m_{11}^{max}$ is stochastically smaller than the distribution of $m_{11}^{sum}$; and for $t = 2$, the density of $m_{11}^{sum}$ is symmetrical with respect to $L/2$.

## 3 Results

### 3.1 Simulated results

Our only option is to simulate the distributions using Monte Carlo methods, which can be done fast under $\bar{H}_0$. We use the *R* software for raw data manipulation, and the *Julia* software [13] to run the hypothesis tests. The data can be found in http://urugenomes.org/lovd/variants, and the *R* and *Julia* code can be found in https://github.com/gabriel-illanes/Ancestors_test.

**3.1.1 Simulated distributions under $\bar{H}_0$.** We compare the effect of increasing the number of generations $t$, and the effect of choosing as statistic the maximum length of $\mathcal{P}_1$-tracts ($m_i^{max}$) or the sum of lengths of $\mathcal{P}_1$-tracts ($m_i^{sum}$). In Fig 2 we show the simulations for the 11<sup>th</sup> chromosome, as it has the mean length of the rest of the chromosomes.

As expected, the statistics decreases as $t$ increases; for $t = 6$ we already observe a very large value of $\mathbb{P}(m^{max} = 0)$. Also, we verify that $M^{max}$ is stochastically smaller than $M^{sum}$, as the sum of lengths will always be larger than the maximum length.

For verifying Equation 2.6 and validate that we can estimate the distribution function of $P_i$ using only the atoms $\omega_0$ and $\omega_L$, we first simulate 10000 values of $m_{11}^{max}$. From Equation 2.6, we observe that we can estimate the distribution function of $P_{11}^{max}$ using only the estimated values of $\omega_0$ and $\omega_L$. Whereas a more naive and inefficient method would be to simulate a new set of 10000 values of $m_{11}^{max}$, obtain a vector of chromosome scores $p_{11}^{max}$ and use them to create the empirical cumulative distribution function (Fig 3).

**3.1.2 Power of the test.** We have four possible variants of the hypothesis test statistic: for each chromosome compute either the maximum length of the $\mathcal{P}_1$-tract or the sum of the lengths of the $\mathcal{P}_1$-tracts and then combine them into a global statistic either as maximum of all $p_i$ or sum of all $p_i$. We asses the power of the hypothesis test in different scenarios, each one of them being a particular case of the alternative hypothesis $H_1$, based on 1000 Monte Carlo simulations for each one.

The first scenario is, for a given $t$, the ancestors $a_1^t, \ldots, a_{2^{t-1}}^t$ have, in average, $2/2^t$ $\mathcal{P}_1$ genetic information, whereas ancestors $a_{2^{t-1}+1}^t, \ldots, a_{2^t}^t$ have $0$ $\mathcal{P}_1$ genetic information. In other words,

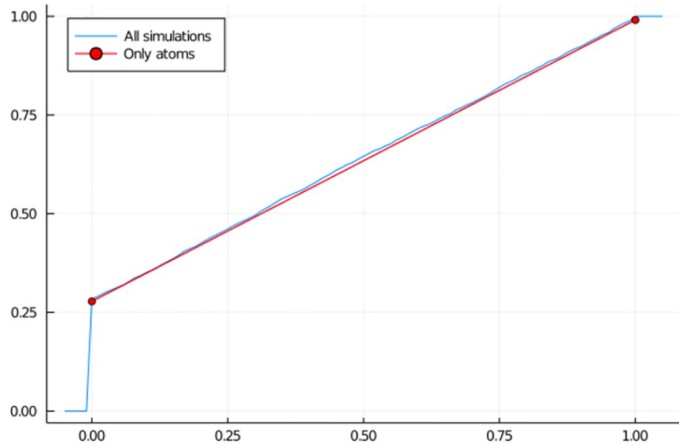

**Fig 3. Empirical cumulative distribution function of the chromosome score $P_{11}$, for chromosome 11 and generation $t = 3$.** In blue, the estimation is done using a new set of 10000 simulations. In red, we use the estimated values of the atoms $\omega_0$ and $\omega_L$ from the original simulations.

we take twice the $\mathcal{P}_1$ genetic information of a $\mathcal{P}_1$-complete ancestor, and spread it across the first $2^{t-1}$ ancestors (the ancestors from one parent's side).

This scenario aims to study the impact of the structure of the $\mathcal{P}_1$-tracts, as we expect to have more $\mathcal{P}_1$ genetic information than in the $\bar{H}_0$ scenario, but resulting from multiple small tracts, rather than several larger ones. We expect that considering the length of maximum $\mathcal{P}_1$-tract as chromosome pair statistic (either $p_{mm}$ or $p_{ms}$) should yield more power for the hypothesis test. The best results are obtained when we consider $p_{mm}$, as expected (Table 1).

The second scenario aims to study what happens when some of the $\mathcal{P}_1$-chromosomes are replaced with $\mathcal{P}_2$-chromosomes. More precisely, we start considering the $\bar{H}_0$ scenario, take the $\mathcal{P}_1$-complete ancestor, and replace half of their chromosomes (in average) with $\mathcal{P}_2$-chromosomes.

As we expect to obtain half of the $\mathcal{P}_1$ genetic information compared to the $\bar{H}_0$ scenario, considering the sum of all the chromosome pair scores as test statistic (either $p_{ms}$ or $p_{ss}$) should yield more power for the hypothesis test. The best results are obtained when we consider $p_{ms}$ and $p_{ss}$, as expected (Table 2).

The third scenario aims to understand what happens, when we do the hypothesis test for generation $t$, but the true scenario is $\bar{H}_0$ for generation $t + 1$ (the first $\mathcal{P}_1$-complete ancestor can be found $t + 1$ generations ago). *A priori*, it is not clear, which of the methods will work best, as we expect smaller $\mathcal{P}_1$-tracts, and less $\mathcal{P}_1$ genetic information than the $\bar{H}_0$ scenario for generation $t$.

**Table 1. Estimated power of the test under the first scenario (ancestors on one parent's side average $2/2^t$ $\mathcal{P}_1$ genetic information, ancestors on father's side average 0 $\mathcal{P}_1$ genetic information) for the four variants of the hypothesis test.**

| Method | $t = 2$ | $t = 3$ | $t = 4$ | $t = 5$ |
|--------|---------|---------|---------|---------|
| $p_{mm}$ | 0.995 | 0.952 | 0.431 | 0.008 |
| $p_{ms}$ | 1.0 | 0.949 | 0.004 | 0.0 |
| $p_{sm}$ | 0.996 | 0.919 | 0.2 | 0.001 |
| $p_{ss}$ | 1.0 | 0.794 | 0.001 | 0.0 |

**Table 2. Estimated power of the test under the second scenario (under $\bar{H}_0$, replace half of a $\mathcal{P}_1$-complete ancestor's chromosomes for $\mathcal{P}_2$-chromosomes) for the four variants of the hypothesis test.**

| Method | $t = 2$ | $t = 3$ | $t = 4$ | $t = 5$ |
|--------|---------|---------|---------|---------|
| $p_{mm}$ | 0.694 | 0.504 | 0.332 | 0.292 |
| $p_{ms}$ | 0.999 | 0.832 | 0.574 | 0.377 |
| $p_{sm}$ | 0.701 | 0.527 | 0.34 | 0.249 |
| $p_{ss}$ | 0.999 | 0.823 | 0.575 | 0.359 |

**Table 3. Estimated power of the test under the third scenario (testing for generation $t$ when simulating for generation $t + 1$) for the four variants of the hypothesis test.**

| Method | $t = 2$ | $t = 3$ | $t = 4$ | $t = 5$ |
|--------|---------|---------|---------|---------|
| $p_{mm}$ | 0.323 | 0.261 | 0.23 | 0.208 |
| $p_{ms}$ | 0.998 | 0.881 | 0.622 | 0.404 |
| $p_{sm}$ | 0.347 | 0.224 | 0.213 | 0.201 |
| $p_{ss}$ | 1.0 | 0.891 | 0.602 | 0.406 |

The best results are obtained when we consider $p_{ms}$ and $p_{ss}$ (Table 3). We could conclude that the impact of obtaining less $\mathcal{P}_1$ genetic information is larger than the length reduction of the $\mathcal{P}_1$-tracts.

From this results several remarks can be done:

- The most impactful decision, under the studied scenarios, is how to combine the chromosome scores to obtain a test $p$-value, rather than how to obtain a chromosome score.

- For almost every scenario and choice of method, we obtained a test power greater than 0.05, which is the expected power under the null hypothesis.

- The difficulty of the problem increases rapidly when increasing $t$. When $t > 4$, we can not expect to obtain reliable results. When $t = 7$, the (simulated) probability of all the $\mathcal{P}_1$ genetic information disappearing (genetic drift) is greater than 0.05, so we would obtain power 0 for any test with level $\alpha = 0.05$.

## 3.2 Empirical results and discussion

We applied the hypothesis test onto a real data set, originated in the context of the project *Urugenomes*. In this project, 10 Uruguayan individuals of known Amerindian ancestry (probably Charrúas) were analyzed; these are the same 10 individuals that were studied in [1]. The inclusion criteria to be part of the study was to have at least one indigenous great grandfather or great great grandfather, according to social anthropological studies and family records and genealogies. Additionally, 10 individuals of known African ancestry were included, that did not know about their Amerindian ancestry. According to historical records, after the first Europeans (Spanish and Portuguese) came to the country, Africans were brought as slaves. Recent results of the Urugenomes project (urugenomes.org) show that these African descendants have also admixture with Amerindian (manuscript in preparation), so they were included in the present study.

Whole genome sequencing of these 20 individuals was done using NGS. Variants were determined and results were phased (haplotype constructions) using 1000Genomes project as reference panel [14]. For this study, we kept only 363578 genome-wide variants, which

correspond to the genotyping array positions used in [15] to study Native American populations. Phased variants were used to construct ancestry specific haplotypes (local ancestry estimations) using RFMix [10]. For this, a reference panel was used that contained complete individuals of European, African and Amerindian ancestries. As a result the data set contained ancestry specific segments of different lengths for each individual. For the purpose of the present work, only indigenous ancestry segments were considered, while the other two ancestries (African and European) were masked out of the data. In summary, the data set is represented by a matrix of 40 haplotypes (corresponding to 20 individuals) and 363578 variants, where information is kept only within indigenous haplotypes and the rest were set to missing data. Out of the matrix, the length distribution of indigenous segments in each haplotype was determined, which is the starting point of the proposed algorithm. For each individual, we obtained their chromosome statistics (either the length of maximum indigenous tract, or the sum of lengths of all indigenous tracts), and undertook all four variants of the hypothesis test. We tested, for $t = 2, \ldots, 5$, whether the individuals might have had at least a complete Amerindian ancestor $t$ generations ago. Fig 4 presents the obtained $p$-values for each individual, method and value of $t$.

We observe that the factor that impacts the most option is the combination of chromosome scores, as we observe larger $p$-values when we consider the sum of all chromosome scores as the test statistic. This can be interpreted as observing more indigenous genetic information than the expected under $\bar{H}_0$, but there are no chromosomes with very large scores. This is a similar behavior as the one observed in the first simulated scenario. That being said, in general, rejecting the hypothesis test for at least one of the statistics should be enough statistical evidence to conclude that the individual does not have any complete Amerindian ancestor $t$ generations ago, specially considering the low power of these tests. In other words, in order to reject the null hypothesis for a given $t$ and a given individual, we should focus on the smallest $p$-value across all statistics.

Considering the test results for the $p_{mm}$ and $p_{sm}$ statistics, we observe that there is a concordance of the test with the expected biological results. Individuals 12, 14, 15, 16, 19 and 20 do not reject the null hypothesis for the presence of a complete Amerindian ancestor for $t = 3$

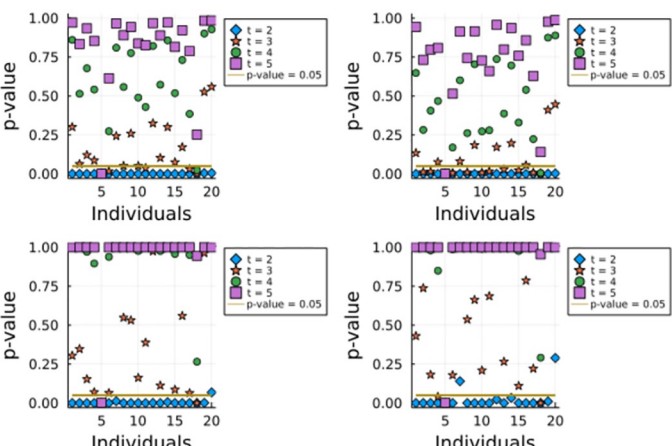

**Fig 4.** Estimated values of $p_{mm}$ (top left), $p_{sm}$ (top right), $p_{ms}$ (bottom left), and $p_{ss}$ (bottom right), using 10000 iterations, for every individual and every $t = 2, \ldots, 5$. Given a choice of statistic, an individual, and fixed $t$, we will reject the null hypothesis if the corresponding $p$-value is below 0.05 (shown in the horizontal line in all graphics). For a unified criterion across all statistics, if one of them rejects the null hypothesis, that is enough statistical evidence to reject $H_0$ for the given individual and $t$.

generations ago -they could have had a complete Amerindian ancestor 3 generations ago-, whereas individuals 11, 13, 17 and 18 reject the null hypothesis for $t = 3$ -there is statistical evidence pointing out that these individuals did not have a complete Amerindian ancestor 3 generations ago-. When considering the test results for $p_{ms}$ and $p_{ss}$, we observe larger $p$-values for every test -across both individuals and generations-, and thus, we do not focus on them. Concurring, individuals 12, 14, 15, 16, 19 and 20 have the largest indigenous ancestry among the 10 individuals with Amerindian ancestry as calculated by genomic approaches.

Regarding the individuals that declared African ancestry (individuals 1 to 10), we observe that they have lower $p$-values, in average, compared to individuals that declared Native American ancestry (individuals 11 to 20). It is important to note that individuals 7 and 9 do not reject the null hypothesis for $t = 3$, but this does not contradict the individuals' declared ancestry (as they could have had at least one complete Native American ancestor 3 generations ago, as well as several complete African ancestors). Another interesting thing to note is that individual 5 rejects the null hypothesis for $t = 5$; it is possible that the individual's family tree members are not originary from South America.

## Supporting information

**S1 File.**
(ZIP)

## Author Contributions

**Conceptualization:** Gabriel Illanes, María Inés Fariello, Ernesto Mordecki, Hugo Naya.

**Data curation:** María Inés Fariello, Lucía Spangenberg.

**Formal analysis:** Gabriel Illanes.

**Investigation:** María Inés Fariello, Lucía Spangenberg, Ernesto Mordecki.

**Methodology:** Gabriel Illanes, Ernesto Mordecki.

**Project administration:** Ernesto Mordecki, Hugo Naya.

**Resources:** Hugo Naya.

**Software:** Gabriel Illanes, María Inés Fariello, Lucía Spangenberg.

**Supervision:** Ernesto Mordecki, Hugo Naya.

**Visualization:** Gabriel Illanes.

**Writing – original draft:** Gabriel Illanes.

**Writing – review & editing:** María Inés Fariello, Lucía Spangenberg.

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
