## [Decision Letter · Decision Letter 0]

7 Apr 2022

PONE-D-21-38583Testing the existence of an unadmixed ancestor from a specific population t generations agoPLOS ONE

Dear Dr. Illanes,

Thank you for submitting your manuscript to PLOS ONE. After careful consideration, we feel that it has merit but does not fully meet PLOS ONE’s publication criteria as it currently stands. Therefore, we invite you to submit a revised version of the manuscript that addresses the points raised during the review process.

Both of the reviewers are unanimous that the study design is good. However, they have raised several concerns, which need a point-by-point reply and amendments in the manuscript. Therefore, please revise the manuscript accordingly.==============================

We look forward to receiving your revised manuscript.

Kind regards,

Gyaneshwer Chaubey

Academic Editor

PLOS ONE

Journal Requirements:

2. Please update your submission to use the PLOS LaTeX template. The template and more information on our requirements for LaTeX submissions can be found at http://journals.plos.org/plosone/s/latex

"Gabriel Illanes acknowledges support of Agencia Nacional de Investigación

e Innivación (ANII-Uruguay) and Comisión Académica de Posgrado (CAP-Udelar)

The Urugenomes project was funded by BID (Banco Iberomericano de desarrollo) Proyecto ATN / KK-L4584-JR “Fortalecimiento de las capacidades técnicas y humanas para las exportaciones de servicios genómicos”.

Additionally, Maria Ines Fariello and Lucia Spangenberg obtained partial

support from the ANII-Uruguay FSDA 1 2017 1 143647 and Lucia Spangenberg and Hugo Naya are also supported by FOCEM (MERCOSUR

Structural Convergence Fund)."

"Gabriel Illanes acknowledges support of Agencia Nacional de Investigaci´on e Innovaci´on (ANIIUruguay) and Comisi´on Acad´emica de Posgrado (CAP-Udelar)

The Urugenomes project was funded by BID (Banco Iberomericano de desarrollo) Proyecto ATN

/ KK-L4584-JR “Fortalecimiento de las capacidades t´ecnicas y humanas para las exportaciones de

servicios gen´omicos”. Additionally, Maria Ines Fariello and Lucia Spangenberg obtained partial

support from the ANII-Uruguay FSDA 1 2017 1 143647 and Lucia Spangenberg and Hugo Naya

are also supported by FOCEM (MERCOSUR Structural Convergence Fund)."

Reviewers' comments:

Reviewer's Responses to Questions

**Comments to the Author**

1. Is the manuscript technically sound, and do the data support the conclusions?

Reviewer #1: Yes

Reviewer #2: Partly

2. Has the statistical analysis been performed appropriately and rigorously? 

Reviewer #1: Yes

Reviewer #2: No

3. Have the authors made all data underlying the findings in their manuscript fully available?

Reviewer #1: No

Reviewer #2: No

4. Is the manuscript presented in an intelligible fashion and written in standard English?

Reviewer #1: Yes

Reviewer #2: Yes

5. Review Comments to the Author

Reviewer #1: Illanes et al, in their present work titled as “Testing the existence of an unadmixed ancestor from a specific population t generations ago” wherein based on ancestry specific segments they hypothesized a method to determine whether an individual has an unadmixed ancestor t generations ago.

Although, the computational modeling and statistical analysis pertaining to the question being asked is satisfactory this manuscript needs some major revision before it is published. Please find the comments below. The page numbers are as per the PDF from journal’s site.

I) Page 2 Section Introduction

1. Third paragraph

“Here, we have developed a hypothesis test to assess whether it is likely that one of the individual’s ancestors t generations ago was an unadmixed ancestor (e.g. complete individuals genome only one ancestry), given a fixed number t of generations and the length of the ancestry-specific tracts for every autosome”

How it is different than

“Determining the Generation at Which an Individual has a Complete Native Ancestor

With High Probability” section in Indigenous Ancestry and Admixture in the Uruguayan Population Spangenberg etal., published in Frontiers in Genetics in September 2021.

There is a need to cite Spangenberg etal. at this point. Also please clarify the additional information the current work is adding to the previously published one.

2. Fourth Paragraph

The line According to historical records, most Uruguayan Amerindian were exterminated in 1831 needs reference.

3. Introduction could be more informative; for example; the authors did not talk about the already available mtDNA and Y-chromosome based analysis on these individuals.

They also did not talk about what already had been established by the previous analysis about the Uruguayan population (admixture, TreeMix etc). All these additions will improve the readability of the manuscript as the current journal targets a wide range of readers.

II) Page 3 Methods

1. In heading add ‘s’ to notation

2. We will assume that a0 and all “her”--- I suggest to write as gender neutral as it might lead to confusion that the analysis might have something to discuss only the maternal ancestry.

3. In Definition 2.4 –For a given λ ∈ Λ, we say that an individual a is Pλ-

complete if all “her” chromosomes are Pλ-tracts

The hypothesis test

……Our strategy, then, is to focus on a " borderline" replace ” before borderline with ” case of H0, where we can fix the ancestors’ pedigree

III) Page 4 Mathematical model

1. The complete essence of the paper relies on this section. So, I strongly recommend a schematic diagram for this section in addition to what authors provided.

Second and the major point is all the four steps in modeling

1. “Recombination………………………….. Li}”

2. “A parent chromosome……………………. daughter chromosome”

3. “At the point x1……………………………. [x1,x2]”

4. “Iterate………………………………….. Li”

Are exactly similar to

“1) Simulate the recombination points using a Poisson process with

parameter Li (length of the chromosome in Morgans). After

adding the borders of the interval [0,Li], we obtain {x0 0, x1,

. . . ,xn,xn+1 Li}. As long as we measure the intervals in

Morgans, using a Poisson process to simulate the

recombination points is not a strong assumption for the model.

2) Select a chromatid at random, and consider the segment tr1

[0,x1] in the selected chromatid. This will be the first segment

of the daughter chromatid.

3) Switch to the other chromatid, and concatenate the segment

tr_2 [x1,x2].

4) Iterate the last step until the length of the daughter chromatid

is Li.” Is already mentioned by Spangenberg et al, 2021

Please do cite and better write that the mathematical model you are following is in concordance with previously published article.

2. Please keep all the figure legends at the end.

3. Remove an extra “.” in legend of Figure 1 “Recombination ……P2-complete (blue). chromosome.”

IV) Page 5 para1

As it possess the Markov property, it is easier to develop mathematical models and tests; however, it fails to capture some structures when we work at an individual level, with small values of t

What structures exactly??

V) In all the figures 3 to 9 please label X and Y axes clearly. Also the legends should be self-explanatory.

VI) References needs to be checked properly,

1. the first reference is already published but it is still citing biorxiv

2. Reference 3rd is incomplete

3. Please refer to the published articles in Plos One for writing journal’s name in reference, instead of full name please use standard scientific Abbreviations

Other comments

1. It would be quite interesting if this model could be tested on some of the other known unadmixed populations and would enhance the strength of the paper.

2. In figures 6-9, please elaborate observation regarding all the individuals e.g individuals 12,14,15,16 and 19 reject hypothesis

Similarly also discuss other individuals especially individual 5

3. While the codes https://github.com/gabriel-illanes/Ancestors_test are publicly available; Variants data unable to be retrieved from the http://urugenomes.org/lovd/variants probably due to some technical. Please ensure the public availability of the data.

Finally, I appreciate the efforts by the team involved in this manuscript. But a thorough revision is definitely needed in the above aspects.

Reviewer #2: The objective of this paper is to determine the timing of ancestry of a particular population, given an individual's genome and panel of reference populations. The paper is generally readable. However, I found it confusing that the empirical analysis is on a group of individuals for whom a reference population is necessarily lacking (because no non-admixed individuals are living), yet a reference population is provided. Some clarity on how the reference panel of data were determined should be provided. Due to this questionable reference comparison, the questionable statistics, and the lack of data availability, I am unsure whether to believe the conclusions in this paper. However, some of this may be solvable with clearer and more comprehensive explanations.

There is no explanation of the simulations other than a reference to code. The code lacks comments that would explain the analysis. Additionally, the code refers to data not contained in the github repo. There's a link to click for the variants but when I do this it says "No variants found" so it's unclear how I would obtain data to run the code so I could make my own attempt to sort out what the code does (which I should not have to do). It's unclear if these were supposed to be simulated data or human genome variants. Either way, I do not see any data.

There should be further details on variant identification including tools, versions, and parameters.

Page 9 states "the length distribution of indigenous segments in each haplotype was determined". Since no non-admixed individuals are known, how was this determined? Who are the complete individuals of Amerindian ancestry?

Figures 6-9 should not be line graphs. These can be combined into a multi panel figure with a much-expanded caption that provides more detailed explanation.

This manuscript has some duplication with Spangenberg et al. BioRxiv 2021 -

https://www.biorxiv.org/content/10.1101/2021.06.09.447750v1.full , which claims to develop the same test described in the manuscript. Assuming that Spangenberg et al. correct their manuscript to state that they only use the test in this manuscript, it is still problematic that both manuscripts describe the empirical results of the test using this particular set of Uruguayan individuals. Given the nature of this manuscript using empirical data from known pedigrees (e.g. for D. melanogaster) would be far more logical.

In the concluding paragraph where does the biological expectation of " a complete Amerindian ancestor only 2 generations ago " come from?

The paper states " Individuals 12, 14, 15, 16, 19 and 20 reject the hypothesis for the presence of a complete Amerindian ancestor t = 5, t = 4 and t = 3 generations ago. " but p values near 0 appear for t=2 for nearly all individuals (although this depends on the test). This would support H1 that there were no complete ancestors at t=0. High p values for other t are shown in the figures. I would have drawn the conclusion that for some individuals you can't reject a complete ancestor at t=3, and for almost all at t=4, and all but 1 at t=5 (at least looking at Fig 6 - since you give four different tests I could use some guidance on which to believe when they conflict).

6. PLOS authors have the option to publish the peer review history of their article (what does this mean?). If published, this will include your full peer review and any attached files.

Reviewer #1: No

Reviewer #2: No

---

## [Author Response · Author response to Decision Letter 0]

23 May 2022

The answer to all questions and comments posed by the Editor and Reviewers can be found in the attached 'Response to Reviewers' file.

---

## [Decision Letter · Decision Letter 1]

24 Jun 2022

Testing the existence of an unadmixed ancestor from a specific population t generations ago

PONE-D-21-38583R1

Dear Dr. Illanes,

We’re pleased to inform you that your manuscript has been judged scientifically suitable for publication and will be formally accepted for publication once it meets all outstanding technical requirements.

Kind regards,

Gyaneshwer Chaubey

Academic Editor

PLOS ONE

Additional Editor Comments (optional):

Reviewers' comments:

Reviewer's Responses to Questions

**Comments to the Author**

1. If the authors have adequately addressed your comments raised in a previous round of review and you feel that this manuscript is now acceptable for publication, you may indicate that here to bypass the “Comments to the Author” section, enter your conflict of interest statement in the “Confidential to Editor” section, and submit your "Accept" recommendation.

Reviewer #1: All comments have been addressed

2. Is the manuscript technically sound, and do the data support the conclusions?

Reviewer #1: Yes

3. Has the statistical analysis been performed appropriately and rigorously? 

Reviewer #1: Yes

4. Have the authors made all data underlying the findings in their manuscript fully available?

Reviewer #1: Yes

5. Is the manuscript presented in an intelligible fashion and written in standard English?

Reviewer #1: Yes

6. Review Comments to the Author

Reviewer #1: Illanes et al, in their present work titled as “Testing the existence of an unadmixed ancestor from a specific population t generations ago” wherein based on ancestry specific segments they hypothesized a method to determine whether an individual has an unadmixed ancestor t generations ago. The authors of the work were suggested to revise the manuscript.

They have addressed the concerns raised satisfactorily. The revised manuscript could be accepted for the publication.

There are few minor issues which the authors could incorporate:

1. Second paragraph page 12:

There are some extra ‘-‘ and ‘.’ Kindly correct the typos.

2. Although in the authors have mentioned

https://filebox.cmat.edu.uy/s/wbRwDHxS8E28m3m for the data in their reply. The final, working and confirmed link for accessing the data by the readers of this journal should be provided.

7. PLOS authors have the option to publish the peer review history of their article (what does this mean?). If published, this will include your full peer review and any attached files.

Reviewer #1: No

---

## [Editor Report · Acceptance letter]

19 Jul 2022

PONE-D-21-38583R1 

Testing the existence of an unadmixed ancestor from a specific population t generations ago 

Dear Dr. Illanes:

I'm pleased to inform you that your manuscript has been deemed suitable for publication in PLOS ONE. Congratulations! Your manuscript is now with our production department. 

Kind regards, 

on behalf of

Gyaneshwer Chaubey 

Academic Editor

PLOS ONE